# Feasibility and Accuracy of Thoracolumbar Pedicle Screw Placement Using an Augmented Reality Head Mounted Device

**DOI:** 10.3390/s22020522

**Published:** 2022-01-11

**Authors:** Henrik Frisk, Eliza Lindqvist, Oscar Persson, Juliane Weinzierl, Linda K. Bruetzel, Paulina Cewe, Gustav Burström, Erik Edström, Adrian Elmi-Terander

**Affiliations:** 1Department of Neurosurgery, Karolinska University Hospital, 171 64 Stockholm, Sweden; oscar.persson.1@ki.se (O.P.); gustav.burstrom@ki.se (G.B.); erik.edstrom.1@ki.se (E.E.); adrian.elmi.terander@ki.se (A.E.-T.); 2Department of Clinical Neuroscience, Karolinska Institutet, 171 77 Stockholm, Sweden; paulina.cewe@ki.se; 3Brainlab AG, Olof-Palme-Str. 9, 81829 München, Germany; eliza.lindqvist@brainlab.com (E.L.); juliane.weinzierl@brainlab.com (J.W.); linda.bruetzel@brainlab.com (L.K.B.); 4Department of Trauma and Musculoskeletal Radiology, Karolinska University Hospital, 171 64 Stockholm, Sweden

**Keywords:** surgical navigation, minimally invasive surgery, spine surgery, augmented reality, pedicle screw, phantom

## Abstract

Background: To investigate the accuracy of augmented reality (AR) navigation using the Magic Leap head mounted device (HMD), pedicle screws were minimally invasively placed in four spine phantoms. Methods: AR navigation provided by a combination of a conventional navigation system integrated with the Magic Leap head mounted device (AR-HMD) was used. Forty-eight screws were planned and inserted into Th11-L4 of the phantoms using the AR-HMD and navigated instruments. Postprocedural CT scans were used to grade the technical (deviation from the plan) and clinical (Gertzbein grade) accuracy of the screws. The time for each screw placement was recorded. Results: The mean deviation between navigation plan and screw position was 1.9 ± 0.7 mm (1.9 [0.3–4.1] mm) at the entry point and 1.4 ± 0.8 mm (1.2 [0.1–3.9] mm) at the screw tip. The angular deviation was 3.0 ± 1.4° (2.7 [0.4–6.2]°) and the mean time for screw placement was 130 ± 55 s (108 [58–437] s). The clinical accuracy was 94% according to the Gertzbein grading scale. Conclusion: The combination of an AR-HMD with a conventional navigation system for accurate minimally invasive screw placement is feasible and can exploit the benefits of AR in the perspective of the surgeon with the reliability of a conventional navigation system.

## 1. Introduction

An increasing number of pedicle screws are placed using minimally invasive techniques [1]. This has the benefit of fewer surgical site infections, less blood loss and shortened hospital stay [2,3,4,5]. The correct placement of pedicle screws is important to prevent vascular and neurological injuries during surgical procedures. While the reported accuracies of freehand pedicle screw placement in the thoracic and lumbar spine are highly variable, the benefits of improved accuracy, especially in minimally invasive surgery, are evident [6,7].

The traditional image guidance method for pedicle screw placement has been intraoperative fluoroscopy, providing two-dimensional information. The continuous development of three-dimensional imaging technologies for intraoperative use has paved the way for surgical navigation in spine surgery. Navigation improves accuracy and reduces the need for revision surgery [7,8]. In addition, it eliminates the need for intraoperative fluoroscopy and consequently the radiation exposure in the OR [9,10,11]. However, surgical navigation carries features that may limit its use. Intraoperative 3D-imaging requires dedicated equipment, and the navigational information must be presented and understood effectively. The latest development in navigated spine surgery is the use of augmented reality (AR) to make navigational information more intuitive by superimposing it on the real world in real-time. AR technology has been shown to provide a meaningful addition in terms of accuracy and usability for navigated pedicle screw placement in the thoracic and lumbar spine [12,13].

An important feature for all navigation systems is the interface used to present the navigational data. While conventional surgical navigation systems typically use screens to provide information in the standardized perspectives common for radiology, AR-solutions are inherently different and lend themselves to presenting information from the perspective of the surgeon [14,15,16,17]. To date, four different interfaces for AR-navigation have been presented [13,18]. Projector based AR was first described in the early 2000s and the first system for spinal navigation was presented in 2020 [19,20,21]. Monitor-based-AR, where a separate monitor displays a video feed of the surgical field combined with AR overlay, has been described in several publications [13]. Microscope-based AR, where the AR image is superimposed on the microscope view as a head up display, has been successfully used in different spinal applications. However, this method is best suited for intra- and extradural tumor surgery and not optimized for pedicle screw placement [22]. Since their introduction, Head-mounted-AR devices have gained attention due to their ability to present information without interrupting the surgeon’s line of sight and causing undue attention shifts [18,23]. Consequently, several head-mounted devices (HMD) using augmented reality navigation are available on the market today. Placing the navigation device on the surgeon’s head allows the augmented reality to be directly projected onto the physical patient and eliminates the need to switch focus between the navigational information and the surgical field. The HoloLens (Microsoft) and XVision (Augmedix) have both been used in studies on pedicle screws placement [24,25,26,27,28,29,30]. A third head mounted device, the Magic Leap, has so far been used in two studies of functional neurosurgery and open spine surgery [31,32].

In this study, we investigate the technical and clinical accuracy of minimally invasive thoracolumbar pedicle screw placement with AR navigation using the Magic Leap head mounted device in combination with a conventional surgical navigation system.

## 2. Materials and Methods

### 2.1. OR Setup, Navigation System and HMD

The study was conducted in a hybrid OR. The hybrid OR was equipped with a motorized ceiling-mounted C-arm system (AlluraClarity Flexmove, Philips Healthcare, Best, The Netherlands) for intraoperative Cone beam CT (CBCT) imaging.

A software prototype designed for AR-navigation using the Magic Leap HMD (Magic Leap 1, Plantation, Magic Leap, FL, USA) was installed on the Curve^®^ 1.0 navigation platform (Brainlab AG, Munich, Germany), and used for all experiments.

### 2.2. Workflow

#### 2.2.1. Co-Registration

Four commercially available spine phantoms for minimally invasive surgery training (Th10-Sacrum) were used (Misstrainer spine surgery: Creaplast, Verton, France). The phantoms consisted of X-ray dense bone substitute, elastic foam soft tissue and silicone skin. For each phantom, a high-resolution CT scan was performed preoperatively (IQon Spectral CT, Philips, Best, The Netherlands). At the beginning of each surgery, the phantom was placed on the OR table and a radiolucent reference clamp with a 4-sphere array was attached to the spinous process of Th10 via a small incision. The universal AIR matrix (Brainlab AG, Munich, Germany) was placed on the phantom and an intraoperative CBCT scan was performed to allow automatic co-registration. The preoperative CT images were fused with the CBCT images for optimal image quality and used by the software to render a 3D-model of the spine. Due to the properties of the phantom, manufactured with the spine in a prone position and with dense foam resisting intervertebral movement, rigid image fusion was adequate to align pre- and intraoperative images.

#### 2.2.2. HDM Calibration and Alignment to Navigation

Each surgeon was fitted with an HMD, which was individually eye calibrated. To align the coordinate systems of the HMD and the navigation system, a custom disposable hybrid marker (Brainlab AG, Munich, Germany) was placed in the field of view of both systems. After alignment, the 3D-model was accurately augmented onto the phantom without the need for additional adjustments. Accuracy was checked using the navigational pointer to touch anatomical landmarks which were accessible at the attachment site of the reference. The accuracy of the overlay of the AR image and the dynamic reference frame was confirmed visually: the reflective spheres of the reference frame were overlayed with green circles when correctly aligned (Figure 1).

#### 2.2.3. Planning and Placing the Screws

Screw planning was performed as part of the intraoperative workflow. A navigated pointer was used to indicate the desired entry and path for each screw on the augmented 3D model (Figure 1).

Each screw placement was assessed and fine-tuned on the touchscreen of the navigation system (Figure 2).

The surgeon incised the skin of the phantom with a scalpel and dissected the foam tissue to reach the pedicle entry point, using a minimally invasive, technique. A navigated drill guide and screwdriver (Expedium, DePuy Synthes, Raynham, MA, USA) were calibrated to the navigation system and for each screw placement positioned along the planned trajectory. The pedicle was drilled with a 4.5 mm diameter drill bit. The navigated screwdriver was then used to place a 5 mm diameter screw in the pedicle (Appendix A). The time (skin to skin) for every screw placement was recorded. Two surgeons planned and placed a total of 48 pedicle screws. In each phantom, 12 screws were placed bilaterally in Th11-L4. Each surgeon planned and placed 24 screws. The outermost (Th10 and L5) levels were excluded. The reason was to avoid dislodging the reference frame attached at Th10 and extreme pedicle angles at L5. When all screws were placed, a new CT scan was performed for measurements of screw placement accuracy.

#### 2.2.4. Technical and Clinical Accuracy Evaluation

To assess technical accuracy, the deviation of each placed screw from its planned path was measured at bone entry and at the tip of the screw. Measurements were performed in 3D by fusing the intraoperative scan, including planned screw paths, with the postoperative scan of the placed screws. The angular deviation was calculated based on these data (Figure 3).

Three independent reviewers assessed the clinical accuracy according to the Gertzbein grading scale; grade 0 (screw within pedicle), grade 1 (breach < 2 mm), grade 2 (breach 2 to < 4 mm), and grade 3 (breach > 4 mm) [33]. Gertzbein grades, 0 and 1 were considered clinically accurate. If there was no consensus between reviewers when grading a screw, the median was used.

### 2.3. Statistical Analysis

Accuracy measurements and screw placement times are expressed as mean ± standard deviation and as medians [min-max]. Pearson’s product-moment correlation was performed for calculating r. Statistical analysis was performed using RStudio (RStudio Team (2016). RStudio: Integrated Development for R. RStudio, Inc., Boston, MA, USA) and a *p*-value less than 0.05 was considered as significant.

## 3. Results

### 3.1. Technical Accuracy

A total of 48 pedicle screws were placed. The accuracy was 1.9 ± 0.7 mm (1.9 [0.3–4.1] mm) and 1.4 ± 0.8 mm (1.2 [0.1–3.9] mm) at the entry point (Figure 4) and at the tip of the screw, respectively (Figure 5). The angular deviation was 3.0 ± 1.4° (2.7 [0.4–6.2]°) (Figure 6). The mean pedicle width was 8.5 ± 2.0 mm and the depth from skin to bone entry was 63.9 ± 10.1 mm.

### 3.2. Time

The mean time for implanting a screw was 130 ± 55 s (108 [58–437] s). The full range of implanting times is shown in Figure 7. There was no correlation between accuracy and time to place the instrument (r = 0.226, *p* = 0.12).

### 3.3. Clinical Accuracy

Of the 48 pedicle screws placed, 45 were rated to be clinically accurate. The grading of the screws is shown in Table 1. Three screws were Gertzbein grade 2 and thus considered inaccurate (Figure 8). The absolute interrater agreement for judging screws as accurate or inaccurate was 92%.

## 4. Discussion

Compared to the conventional free-hand technique, computer-assisted surgery, including AR and robotic applications, has been shown to improve pedicle screw placement accuracy. Projector based AR has been used to reach high clinical accuracy in a cadaveric model [21]. Monitor-based-AR has gradually been improved with increasing clinical accuracy from 85% in the first preclinical study to 94% in challenging clinical deformity cases [14,15], while the technical accuracy of the system has improved from 2.2 ± 1.3 mm using only instrument tracking to 0.94 ± 0.59 when combined with robotic assistance [34,35]. Microscope-based -AR has been used for spinal tumor surgery reaching a target registration error of 0.87 ± 0.28 mm [22]. HMD-AR devices are the latest addition to the AR-navigation arsenal. Studies on non-medical devices (HoloLens, Magic Leap) and FDA-approved medical devices (X-vision) are very promising. The latter has been used in clinical open lumbar surgeries achieving a clinical accuracy of 97.8% [23]. However, the main advantage of AR devices in general, and HMDs in particular, is in assisting in minimally invasive spine surgery.

This is the first study using an augmented reality head mounted device (AR-HMD) in conjunction with an established conventional navigation system for minimally invasive spinal pedicle screw placement. The AR-HMD device gives visual guidance for identifying the bone entry point and directing the instruments during minimally invasive procedures. Locating the bone entry point is a key step. The 3D AR model allows for visualization of the instrument position in respect to the deep bony anatomy, simplifying adherence to the planned path.

The conventional navigation system provides, according to the manufacturer, an overall accuracy of ≤2.0 mm/2° (mean) and ≤3.00 mm/3° (95th percentile). In this prototypic design, the HMD is aligned to the conventional system. The AR-perspective allows the identification of misalignment, which can be managed with a realignment procedure (Figure 9). The AR-HMD device in this study demonstrates a clinical accuracy of 94% and a technical accuracy of 1.9 ± 0.7 mm, which is an excellent performance in light of the inbound error of the system and in line with previously reported data by other AR navigation systems [12,13,15,27,36]. Three screws were graded as Gertzbein grade 2. The preoperative plan and postoperative scans for these three screws are presented in Figure 6, it is evident that all three screws deviate slightly lateral, but they are clinically acceptable and would not require revision or replacement. A screw placed with the “in-out-in” technique although clinically safe, strictly following the Gertzbein scale is considered as inaccurately placed.

Liebmann et al. performed a study where they used the HoloLens to place pedicle screws in an open lumbar spine model and reported 2.77 ± 1.46 mm mean deviation from the planned path at the entry point and an angular deviation of 3.38°± 1.73° [25]. When Molina et al. used Augmedics’ XVision to place thoracolumbar pedicle screws in cadaveric torsos, in open surgery, they achieved a clinical accuracy of 94.6% according to the Gertzbein scale [24]. Yanni et al., used the same AR-HMD in combination with conventional navigation to place pedicle screws in a 3D printed, fully visible, open saw bone model and reached 98.4% accuracy according to the Gertzbein scale without reporting on technical accuracy [32]. In a minimally invasive set-up Liu et al., achieved an accuracy of 94% using Hololens [28].

Accuracy in this study is limited by the intrinsic properties of the HMD-system. In contrast to the solutions studied by Liebmann and Molina, the system evaluated in this study is a combination of outside-in tracking afforded by the underlying conventional reference frame-based navigation system and inside out tracking from the HMD. This combination allows realignment of the AR-view to the conventional tracking when necessary. Of importance, both the technical and clinical accuracy reported in this study reflect the compounded errors of the navigational system and the surgeon’s adherence to the plan. As mentioned above, the target registration error reported for conventional navigation systems is within 1–2 mm. The use of robotic arms may reduce errors related to the surgeon’s ability to maintain the correct position and angle of approach on the bone entry point, thereby increasing the technical accuracy [35].

### 4.1. Attention Shift

The use of an AR-HMD has the benefit of placing the navigation information directly in the surgeon’s field of view. Both 2D and 3D models of the spine are visible with the 3D model being aligned with the phantom’s anatomy. Having the same navigation information directly in the surgical field minimizes the risk of attention shift. Attention shift occurs when surgeons must change their focus repeatedly between the surgical field and the screen of the navigation system to follow the navigational cues. This shift of attention has shown negative effects on motor and cognitive tasks during surgery and could be a source of instrumentation error in spinal surgery [37].

However, the use of HMD limits the information to those wearing them. Thus, a screen in the OR with a live navigation feed is beneficial for the rest of the surgical team, especially the OR nurse. In this study, the navigation screen simplified intraoperative communication, giving the OR nurse information about the surgical procedure. This may reduce surgical flow disruption. Lack of communication in the OR has been reported as a major cause of disruption resulting in prolonged operating time [38,39].

### 4.2. Radiation

Fluoroscopy is traditionally used for intraoperative guidance when placing screws in minimally invasive procedures. Fluoroscopy assisted screw insertion accuracy is reported to range from 83.9% to 100% but is associated with exposing the surgical team to varying amounts of radiation [40,41,42]. Using modern imaging techniques and computer assisted navigation can eliminate the need for intraoperative fluoroscopy and decrease the radiation exposure to the surgical team [43]. No lead aprons were worn during the procedures in this study. Instead, distancing and protective lead shields were used to ensure radiation safety during the initial and final CBCT-scans [44].

### 4.3. Limitations

Limitations of this study include the small sample size and the fact that plastic torso phantoms were used. There were three screws deemed inaccurate in our study. Analysis of the pre-procedural plan and post-procedural images shows that all three screws were correctly planned but deviated slightly laterally. However, these screws would not necessarily have breached if placed in real bone. The plastic used in our model lacked differentiation between cancellous and cortical bone, meaning the typical resistance offered by cortical bone was absent. Moreover, navigation systems based on dynamic reference frames have been shown to lose accuracy with distance from the reference [45]. This is explained by the mobility of the spine, where undetected movements can occur between the vertebra to which the reference is attached, and the vertebra being manipulated surgically. However, no such effect could be seen in this study. A possible explanation for this lies in the properties of the phantom used, where the foam could withstand manipulations of the contained spine model.

These are problems inherent to the study design and future studies on cadavers or animal models, and eventually, clinical trials will allow a better understanding of the efficacy of the system.

## 5. Conclusions

The combination of an HMD with a conventional navigation system for accurate minimally invasive pedicle screw placement is feasible and can exploit the benefits of AR from the perspective of the surgeon with the reliability of a conventional navigation system.

## Figures and Tables

**Figure 1 sensors-22-00522-f001:**
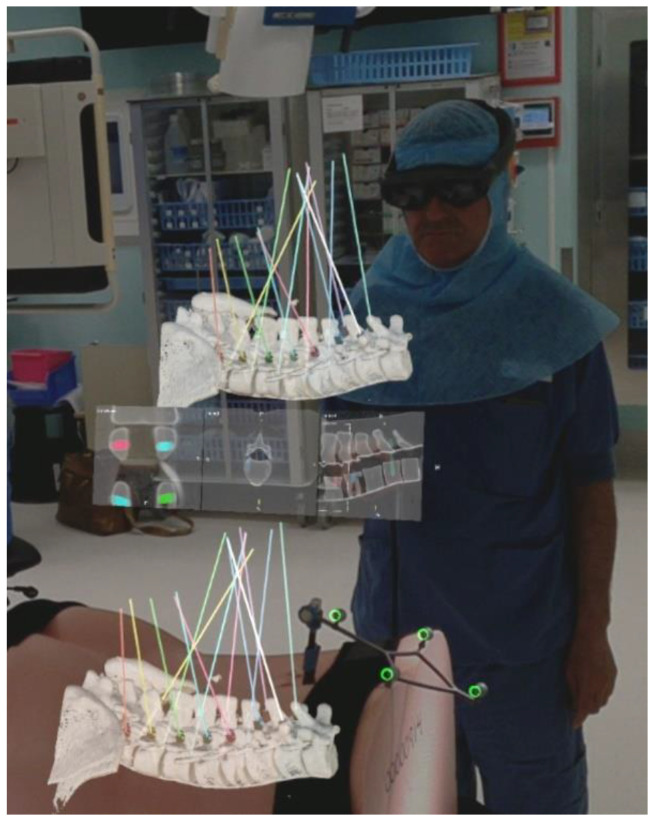
Surgeon’s view through the HMD presenting 2D and 3D information. The 3D overlay contains the planned 3D screws with protruding trajectory lines to assist instrument alignment. The lower 3D model is augmented onto the spine of the phantom while the 2D and 3D-representations seen floating above the phantom provide additional information and can be positioned freely in the virtual space and switched on and off. Additionally, note the colocalization of the green circles with the reflective spheres of the dynamic reference frame indicating accurate alignment.

**Figure 2 sensors-22-00522-f002:**
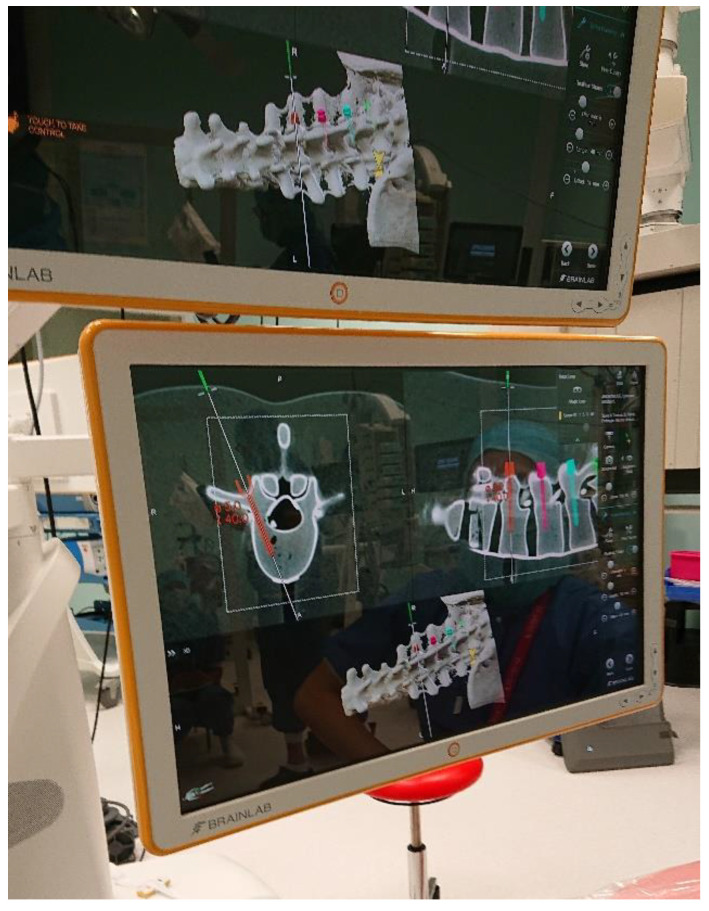
The screens of the navigation system present information corresponding to the surgeon’s view in the HMD.

**Figure 3 sensors-22-00522-f003:**
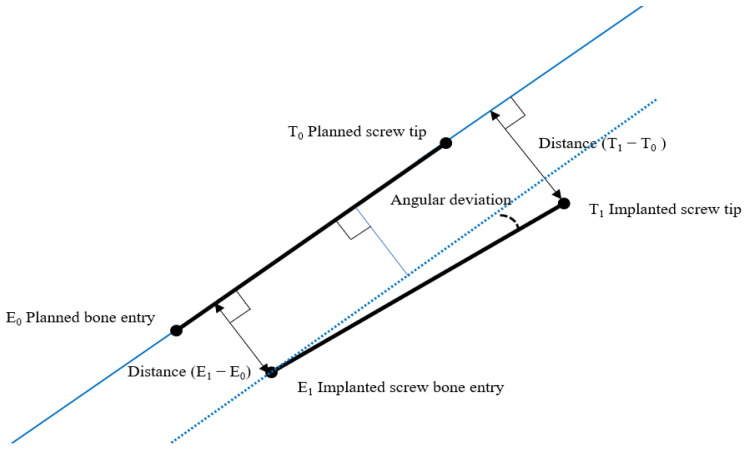
Overview of measurement model for technical accuracy.

**Figure 4 sensors-22-00522-f004:**
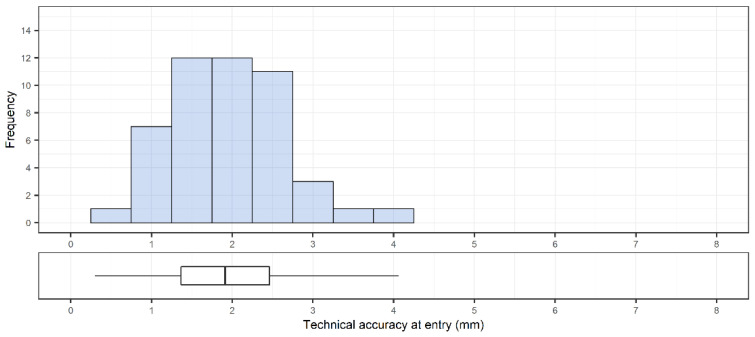
Overall distribution of technical accuracy of placed pedicle screws (n = 48) at bone entry point. Corresponding box plot with median and interquartile range are included below the histogram.

**Figure 5 sensors-22-00522-f005:**
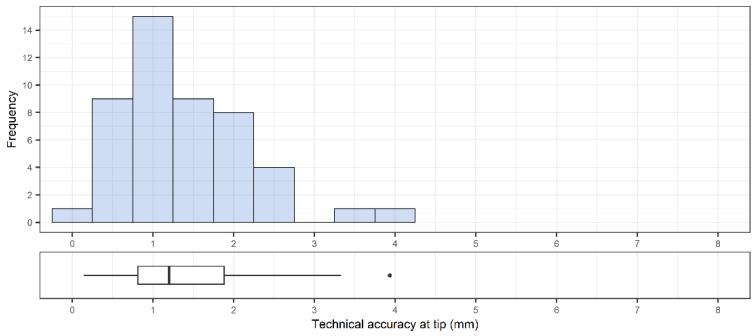
Overall distribution of technical accuracy of placed pedicle screws (n = 48) at screw tip. Corresponding box plot with median, interquartile range and outliers are included below the histogram.

**Figure 6 sensors-22-00522-f006:**
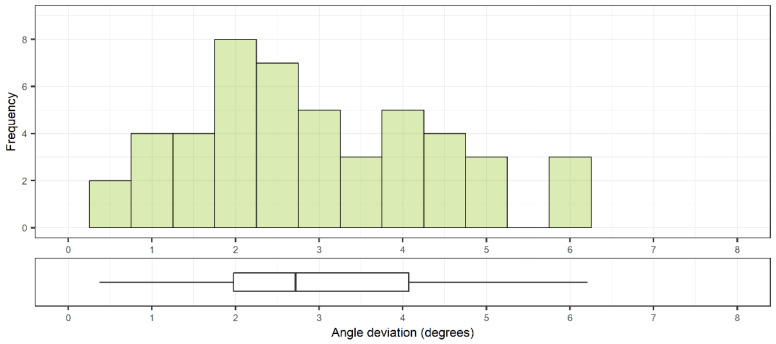
Distribution of angular deviations between planned trajectories and placed screws (n = 48) with corresponding box plot indicating median and interquartile range.

**Figure 7 sensors-22-00522-f007:**
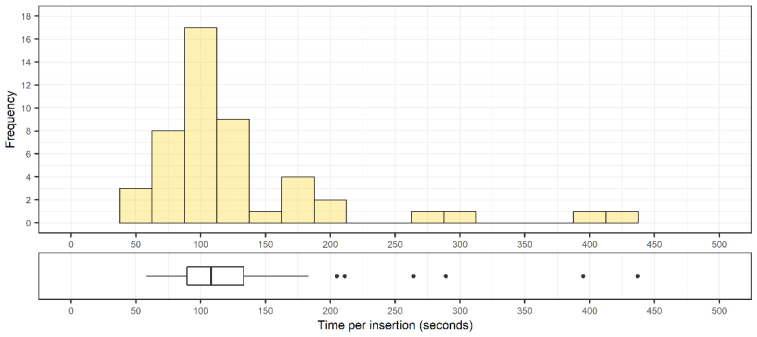
Distribution of screw implantation times (n = 48) with corresponding box plots indicating median, interquartile range, and outliers from the distribution.

**Figure 8 sensors-22-00522-f008:**
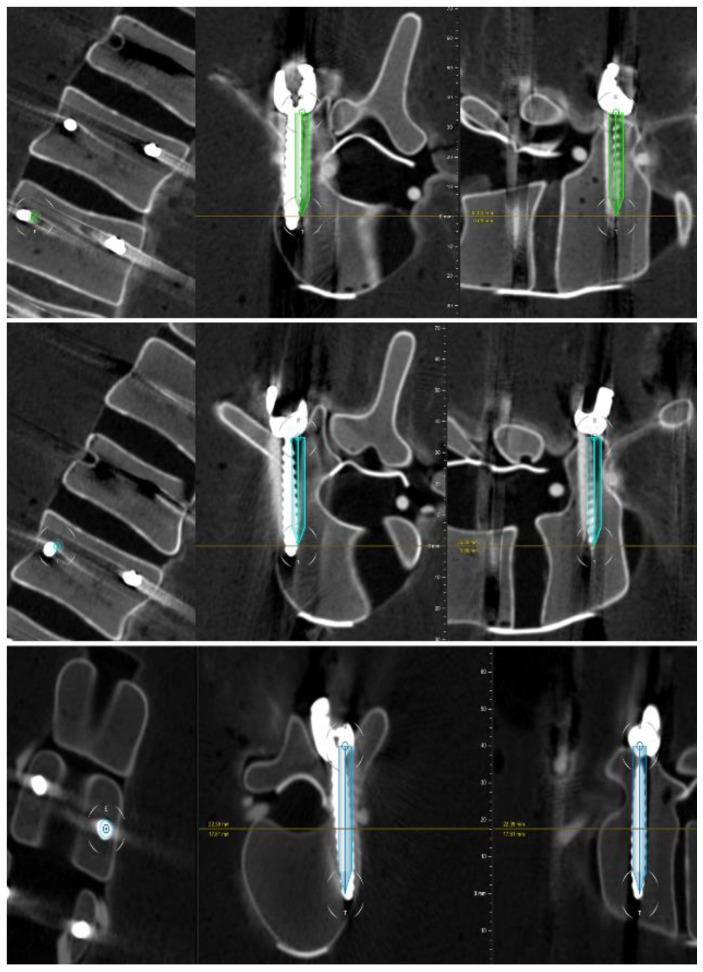
The postoperative scans including the surgical plan for the 3 screws graded 2 on the Gertzbein scale, presented in coronal, axial and sagittal views.

**Figure 9 sensors-22-00522-f009:**
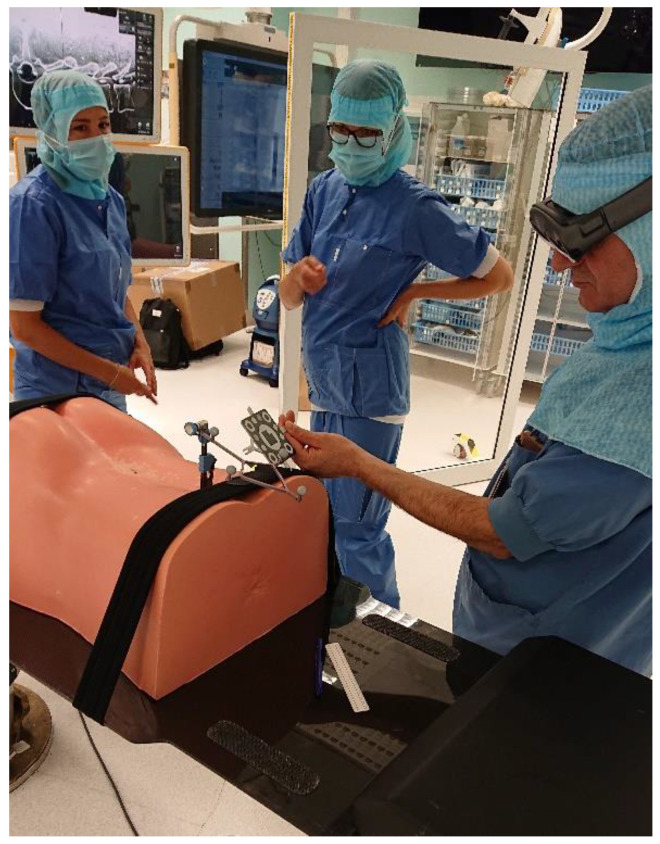
The surgeon calibrating the HMD to the navigation system using a hybrid marker.

**Table 1 sensors-22-00522-t001:** Technical accuracy of implanted screws.

	Gertzbein Grade 0	Gertzbein Grade 1	Gertzbein Grade 2	Gertzbein Grade 3	Clinically Accurate	Clinically Inaccurate	Accuracy
Number of screws	35	10	3	0	45	3	94%

## Data Availability

Data is available from the corresponding author upon reasonable request.

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
