# Peer review of "Feasibility and Accuracy of Thoracolumbar Pedicle Screw Placement Using an Augmented Reality Head Mounted Device"

_sensors, 2022, doi:10.3390/s22020522_

Round 1

Reviewer 1 Report

In this manuscript, the authors evaluated the accuracy of augmented reality (AR) navigation using the Magic Leap head mounted device (HMD) for thoracolumbar pedicle screw placement.

The authors could compare in more details the technology of AR they used with the other AR navigation that have been reported in the literature so far.

For example:

J Neurosurg Spine. 2021 Oct 8;1-7. doi: 10.3171/2021.2.SPINE202097. Online ahead of print. Clinical accuracy and initial experience with augmented reality-assisted pedicle screw placement: the first 205 screws Ann Liu 1, Yike Jin 1, Ethan Cottrill 1, Majid Khan 2, Erick Westbroek 1, Jeff Ehresman 1, Zach Pennington 1, Sheng-Fu L Lo 1, Daniel M Sciubba 1, Camilo A Molina 3, Timothy F Witham 1

In addition, the authors could briefly compare the pros and cons of the AR for pedicle screw placement versus the other techniques used (e.g., fluoroscopy, navigation, robotic arm)

Reviewer 2 Report

I have no comments for Authors. The paper is clearly written, all problems are well described and commented.
Maybe in future studies it will be fruitfull to cooperate with other medical centers and to compare obtained results using similar methodology with apllication of conclusions reached in the present paper.

Reviewer 3 Report

Authors present a phantom study with MIS placement of pedicle screws using AR navigation provided by a combination of a conventional navigation system integrated with the Magic Leap head mounted device (AR-HMD). 48 screws were implanted in 4 phantoms with prior screw planning inTh11-L4; the mean screw deviation was reported to be 1.9±0.7 mm at entry 
point and 1.4±0.8 mm  at the screw tip. The angular deviation was 3.0±1.4°  and the mean time for screw placement was 130 ± 55 seconds, with 94% of correct screws according to GRScale. 

Several important isssues to be adressed:

a) Reference array was attached to Th10; however, in real life, to perform a navigated screw insertion into for example L4 with a reference array which is far away - on Th10 - is usually not a good idea. Where there any differences between PS accuracy between segments which are further away from the array compared to those which are nearby?

b) How was registration accuracy checked?

c)  Did you perform elastic or rigid fusion of preoperative and intraoperative CT? Please provide a figure which shows this.

d) For a reader which has not been familiar with AR, Figure 1. is unclear - does a user see this model of the spine while wearing Magic Leap all the time, in all directions, or only when directed to the structures? Please clarify and provide an operative video or surgeons view photos while implanting screws.

e)Please clarify the technical accuracy, i.e. screw offset - Euclidian distance? 

f) Clarify the high rate of misplaced screws and possible reasons

For Discussion I recommend to include following study:

  1. Carl B, Bopp M, Saß B, Pojskic M, Voellger B, Nimsky C. Spine Surgery Supported by Augmented Reality. Global Spine J. 2020 Apr;10(2 Suppl):41S-55S. doi: 10.1177/2192568219868217. Epub 2020 May 28. PMID: 32528805; PMCID: PMC7263340.
  2. Liu A, Jin Y, Cottrill E, Khan M, Westbroek E, Ehresman J, Pennington Z, Lo SL, Sciubba DM, Molina CA, Witham TF. Clinical accuracy and initial experience with augmented reality-assisted pedicle screw placement: the first 205 screws. J Neurosurg Spine. 2021 Oct 8:1-7. doi: 10.3171/2021.2.SPINE202097. Epub ahead of print. PMID: 34624854.
  3. Uddin SA, Hanna G, Ross L, Molina C, Urakov T, Johnson P, Kim T, Drazin D. Augmented Reality in Spinal Surgery: Highlights From Augmented Reality Lectures at the Emerging Technologies Annual Meetings. Cureus. 2021 Oct 31;13(10):e19165. doi: 10.7759/cureus.19165. PMID: 34873508; PMCID: PMC8631483.

Round 2

Reviewer 1 Report

The authors replied to my comments. I believe the manuscript is improved.

Reviewer 3 Report

Authors have sufficiently repplied to all remarks.